METHODS AND RESOURCES

# Accurate classification of major brain cell types using in vivo imaging and neural network processing

**Amrita Das Gupta[1], Livia Asan[1¤], Jennifer John[1], Carlo Beretta[1], Thomas Kuner[1] \*, Johannes Knabbe**  [1,2] \*

**1** Department of Functional Neuroanatomy, Institute for Anatomy and Cell Biology, Heidelberg University, Heidelberg, Germany, **2** Department of General Psychiatry, Centre for Psychosocial Medicine, Heidelberg University, Heidelberg, Germany

¤ Current address: Department of Neurology and Center for Translational Neuro- and Behavioral Sciences (C-TNBS), University Hospital Essen, Essen, Germany

\* thomas.kuner@uni-heidelberg.de (TK); johannes.knabbe@uni-heidelberg.de (JK)

## Abstract

Comprehensive analysis of tissue cell type composition using microscopic techniques has primarily been confined to ex vivo approaches. Here, we introduce NuCLear (Nucleus-instructed tissue composition using deep learning), an approach combining in vivo two-photon imaging of histone 2B-eGFP-labeled cell nuclei with subsequent deep learning-based identification of cell types from structural features of the respective cell nuclei. Using NuCLear, we were able to classify almost all cells per imaging volume in the secondary motor cortex of the mouse brain (0.25 mm$^3$ containing approximately 25,000 cells) and to identify their position in 3D space in a noninvasive manner using only a single label throughout multiple imaging sessions. Twelve weeks after baseline, cell numbers did not change yet astrocytic nuclei significantly decreased in size. NuCLear opens a window to study changes in relative density and location of different cell types in the brains of individual mice over extended time periods, enabling comprehensive studies of changes in cell type composition in physiological and pathophysiological conditions.

## Introduction

Understanding the plasticity and interaction of different brain cell types in vivo for the investigation of large-scale structural brain alterations associated with diverse physiological and pathophysiological states has been challenging due to the inability to study multiple cell types simultaneously. Until now, to investigate the effect of an experimental intervention on different cortical cell types and their respective spatial interactions, each would require an individual labeling strategy using a set of specific promoters and fluorophores. This would require prioritizing cell types from the outset and thereby limit the scope to a small subset of the population. With multiple cell types to consider, the number of experiments and animals investigated may become intractable. To date, quantitative studies assessing whole tissue composition have

**Data Availability Statement:** All datasets used in the study are available on the heiData archive: https://heidata.uni-heidelberg.de/dataset.xhtml?persistentId=doi:10.11588/data/L3PITA Code and

exemplary data can be found on the Github page: https://github.com/adgpta/NuCLear We also published all 3 Github repositories on Zenodo: NuCLear: https://zenodo.org/badge/latestdoi/662776296 NucleusAI: https://zenodo.org/badge/latestdoi/671099550 SynthGen: https://zenodo.org/badge/latestdoi/665629516.

**Funding:** We gratefully acknowledge the support by the German Research Foundation (DFG) (SFB1158, project B08 awarded to TK), the data storage service SDS@hd, supported by the Ministry of Science, Research and the Arts Baden-Württemberg (MWK) and the DFG through grant INST 35/1314-1 FUGG, as well as the high-performance cluster bwForCluster MLS&WISO, supported by the MWK and the DFG through Grant INST 35/1134-1 FUGG. The funders had no role in study design, data collection and analysis, decision to publish, or preparation of the manuscript.

**Abbreviations:** EEC, European Economic Community; eGFP, enhanced green fluorescent protein; FN, false negative; FP, false positive; H2B-eGFP, histone 2B-eGFP; i.p., intraperitoneal; NRR, normal rat ringer; NuCLear, nucleus-instructed tissue composition using deep learning; ReLU, rectified linear unit; s.c., subcutaneous; SDV, synthetic data vault; TP, true positive.

primarily employed ex vivo approaches using manual or automated analysis of antibody staining in tissue sections [1], isotropic fractionation [2], in situ hybridization [3], or single-cell sequencing [4]. Other studies used manual [5] or automated stereological approaches [6,7]. All these studies were conducted ex vivo and a subset required the dissolution of the studied organ's cellular architecture. The ability to identify multiple cell types and their location in 3D space within a single subject, here referred to as "tissue composition," in the integrity of the living brain to quantify and observe changes over time delivers new opportunities.

Here, we propose an experimental approach to study excitatory and inhibitory neurons as well as glial cells, especially, astroglia, microglia and oligodendroglia, and endothelial cells in the same mouse in vivo over time, using a single genetically modified mouse line expressing a fusion protein of histone 2B and enhanced green fluorescent protein (eGFP) in all cell nuclei. Our approach uses a deep learning method, implementing artificial neural networks to classify each nucleus belonging to a specific cell type, making it possible to perform a nucleus-instructed tissue composition analysis using deep learning (NuCLear). Additionally, determining the precise coordinates of every nucleus within the imaged space allows one to assess spatial relations of cells, e.g., the degree of cell clustering versus even distribution of cells. This can be utilized as an indirect marker of glial territory size, which has been proven to be relevant in pathogenesis of disease such as Alzheimer's [8], or give information about neuron-glia-vasculature proximity [9]. Apart from the application in longitudinal in vivo imaging of the mouse brain, the concept of this technique could be applicable to many other cellular imaging techniques such as confocal and widefield imaging of different organs, when ground truth data for classifier training is available. This will make NuCLear a powerful approach for future analyses of large-scale automated tissue composition.

## Results

For the comprehensive identification and tracking of cell type composition in the mouse brain over the course of weeks and even months, a mouse line constitutively expressing a human histone 2B-eGFP (H2B-eGFP) fusion protein in all cell nuclei was imaged using two-photon microscopy after implantation of a chronic cranial window (Figs 1A and S1A). Volumetric images taken from this mouse line show a nuclei distribution resembling the well-known DAPI staining (S1B and S1C Fig). The idea behind the approach was to use nuclei as proxy for cells and train a neuronal network with ground truth data to classify nuclei belonging to distinct cell types. In brief, the proposed method consists of 3 main steps: (1) automated nuclei segmentation in the raw data (detection and labeling of nuclei); (2) feature extraction of the segmented nuclei; and (3) classification of individual nuclei using pretrained classifiers for each desired cell type (Fig 1B, 1C and 1C(i)). In our study, we selected 5 different cell types for classification: neurons, astroglia, microglia, oligodendroglia, and endothelial cells. Neurons were further classified into inhibitory or excitatory subtypes.

To be able to train a classifier for each cell type, ground truth data had to be generated. In addition to the H2B-eGFP-labeled nuclei, a red fluorescent marker protein was introduced (tdTomato, DyLight 549, or mCherry) to mark a specific cell type. The colocalization of both the red and green label allowed us to assign each nucleus to a specific cell type and to extract ground truth data for nucleus classification in the green channel. Nuclei belonging to a specific cell type were manually selected and their features were used to train a neuronal network classifier in a supervised way (Fig 2A). To avoid a bleed through effect of the red fluorescent protein tdTomato into the eGFP channel, which could influence the quality of the classification, we induced the Cre-ERT2-dependent expression of tdTomato via intraperitoneal injection of tamoxifen (S1D Fig, right) for cell type identification after the first imaging of the nuclei (Fig

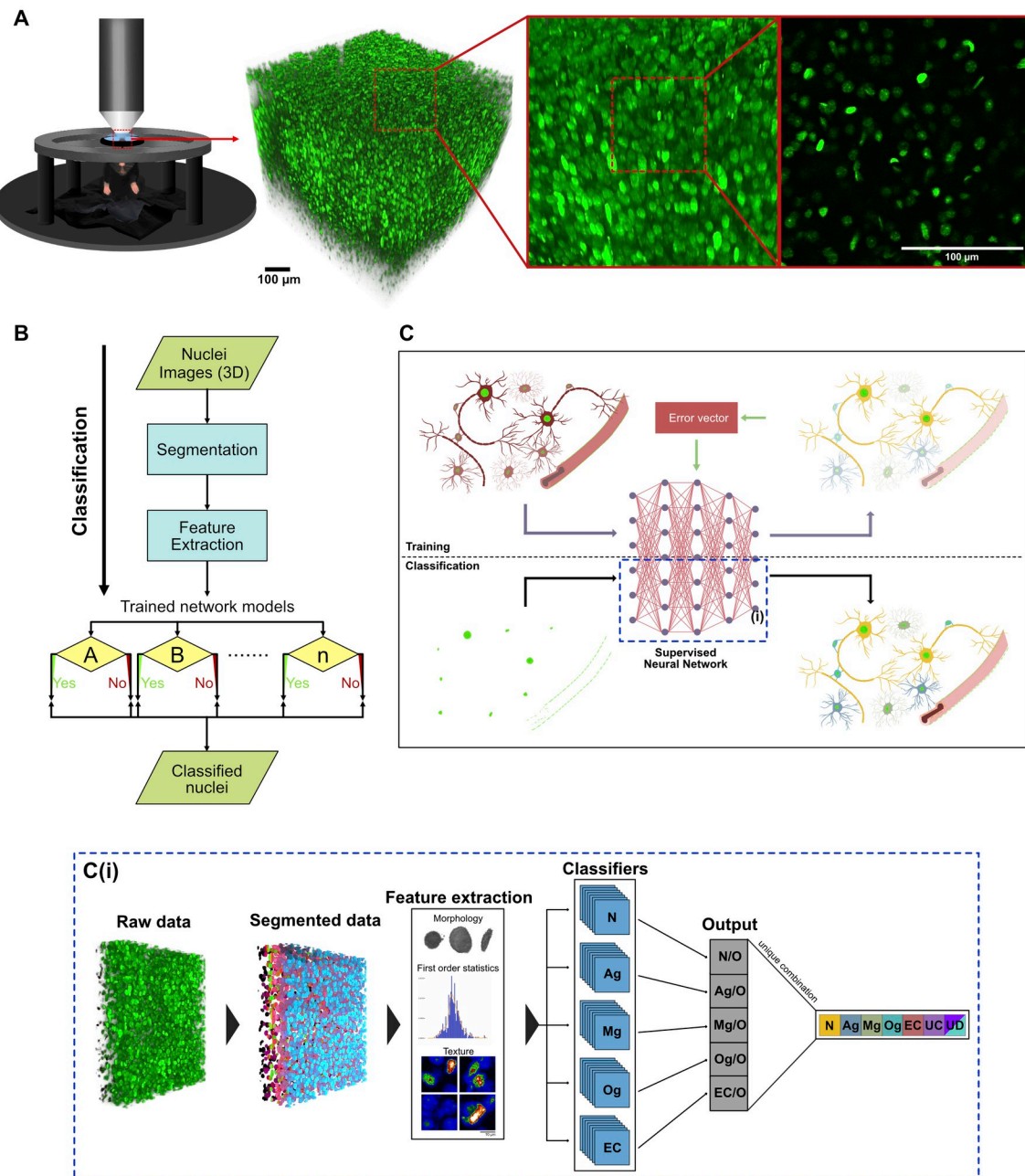

**Fig 1. Imaging of cell nuclei and classification into different types. (A)** Illustration showing acquisition of a
700 μm × 700 μm × 700 μm image volume of cell nuclei in the neocortex using in vivo 2P microscopy in a H2B-eGFP mouse; 3D
reconstruction and single imaging plane. **(B)** Flow diagram showing the classification process from acquisition of the 2P images until
the final classification. **(C)** Illustration visualizing a simplified process of the training and classification algorithms. Using an overlay of
red (only 1 cell type labeled per mouse) and green (nuclei with H2B-eGFP expression: bright green) fluorescent images, ground truth
data were obtained to train a supervised neural network to classify nuclei into cell types. Yellow: Neurons, blue: Astroglia, green:
Microglia, turquoise: Oligodendroglia, red: Endothelial cells. **(C(i))** Visualization of the classification process described in (B) and the
blue dashed area in (C). Image volumes were obtained from H2B-eGFP mice, each nucleus was automatically segmented. Features were
extracted and each nucleus was classified using pretrained classifiers. (N = Neuron, Ag = Astroglia, Mg = Microglia,
Og = Oligodendroglia, EC = Endothelial cells, UC = unclassified cells, UD: undecided cells (classified as belonging to multiple classes)).
2P, 2-photon; H2B-eGFP, histone 2B-eGFP.

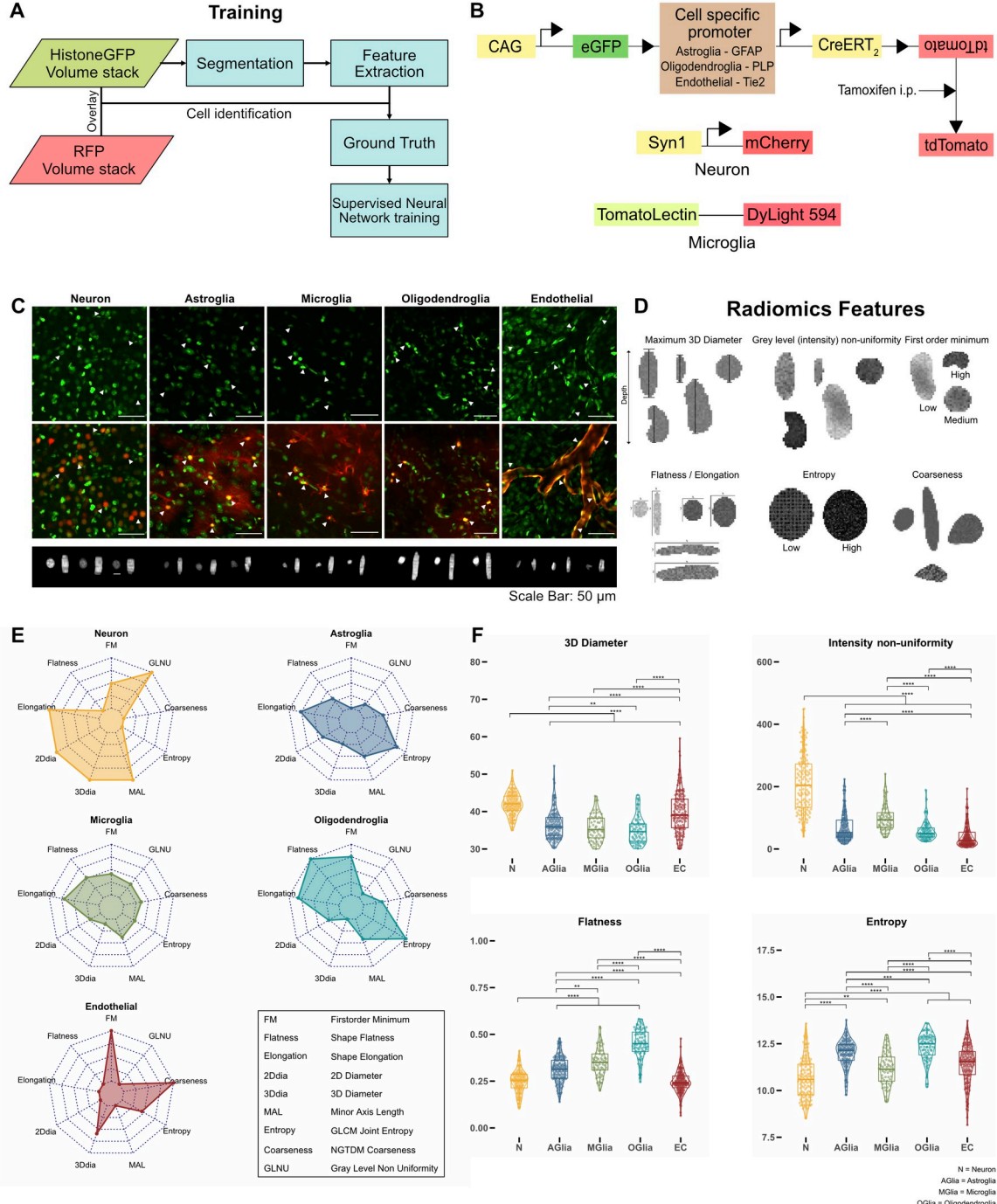

**Fig 2. Strategies for training a neural network classifier using nucleus features.** **(A)** Flow diagram showing classification training pipeline. Cell type specific red fluorescent proteins were used to manually identify H2B-eGFP nuclei. This information was used to train a classifier for each cell type. **(B)** Labeling strategies used for each cell type. Reporter mouse lines were created by breeding H2B-eGFP mice to carry a floxed sequence of the red fluorescence protein tdTomato and a tamoxifen inducible Cre-recombinase under the expression of different cell type specific promoters that were used for identification of astroglia, oligodendroglia, and endothelial cells (see Methods). Microglia and neurons were visualized using intracortical injections of Lycopersicon Esculentum (Tomato) lectin and an AAV expressing mCherry under the synapsin promoter, respectively. **(C)** Nuclei belonging to a specific cell type could be identified by a red fluorescent marker (maximum intensity projections; z = 20 μm). 3D-renderings of individual nuclei are shown in the lower panel. Scale bar: 50 μm. **(D)** Illustrations of a subset of radiomics features showing examples of different shape, intensity, and texture features. **(E)** Radar plots showing a subset of nuclear radiomics features for each cell type. **(F)** Comparison of nuclear features (3D

diameter in voxels, (voxel resolution in xyz: 0.29 μm × 0.29 μm x 2μm), intensity non uniformity, flatness, entropy) between cell types (Wilcoxon test, Bonferroni correction for multiple comparisons, *n* of neurons: 135, *n* of astroglia: 137, *n* of microglia: 62, *n* of oligodendroglia: 72, *n* of endothelial cells: 155, $p < 0.05$ *, $p < 0.01$ **, $p < 0.001$ ***), N = Neuron, AGlia = Astroglia, MGlia = Microglia, OGlia = Oligodendroglia, EC = Endothelial cells. Plot data can be found in S1 Data. H2B-eGFP, histone 2B-eGFP.

2B, overlapping emission spectra for eGFP and tdTomato are shown in S1E Fig, bleed through of the tdTomato signal into the eGFP channel is demonstrated in S1F Fig). As microglia were the only cells to change their positions after induction with tamoxifen, we chose acute injection of tomato-lectin coupled to DyLight 594 to label these cells, which did not produce any signal alterations in the eGFP channel (S1G Fig). Neurons were labeled using a cortically injected AAV expressing the mCherry red fluorescent protein under the neuron-specific synapsin pro-motor (syn1-mCherry), which did not affect the eGFP signal due to its specific fluorescence properties with a more red-shifted emission spectrum compared to tdTomato (S1E and S1G Fig). To further distinguish between different neuronal subtypes, excitatory and inhibitory neurons were labeled by AAV-mediated expression of CamKIIα-mCherry and mDLX-mRuby2, respectively (S2A and S2B Fig). To prepare the ground truth datasets of tdTomato expressing mice, nuclei were manually selected from the preinduction time point images (green fluorescence signal) wherever they overlapped with tdTomato signal in the post-induc-tion time point images (red fluorescence signal) (Figs 2C and S4B). For the microglial and neu-ronal datasets, nuclei that overlapped in the green and red channel were manually selected (Figs 2C and S2B).

Automated segmentation of nuclei in the raw data was achieved using the StarDist neural network [10], trained on manually traced, and labeled ground truth datasets of the H2B-eGFP mouse line (S1H(i)–S1H(ii) and S1I Fig). The trained network showed a high nucleus detec-tion accuracy of 94% as well as a good shape segmentation of nuclei (S1H(iii)–S1H(iv) and S1I Fig). From the binary mask of individual nuclei and their respective pixel intensities in the raw image, in total 107 features were extracted using the PyRadiomics package [11], including 3D-diameter, flatness, gray level non-uniformity, entropy, first order minimum, or coarseness (Fig 2D) (for a full list and short description of the features, see S1 and S2 Tables). To reduce possi-ble overfitting of the classification algorithm, a subset of 12 features were automatically selected using a sequential feature selection algorithm (S2 Table). Radar plots with these nuclear features visualize the differences between the cell types, for example, neuronal nuclei having a larger 3D diameter and minor axis length in comparison to microglial and astroglial nuclei that exhibit a larger pixel entropy (Fig 2E). For certain features, significant differences between the cell types can be shown as well, for example, flatness being significantly higher in nuclei of microglia than in nuclei of neurons (Fig 2F). Furthermore, combinations of 2 or 3 distinctive features allow for the visual separation of nuclei of different cell types in the 2D and 3D space (S3 Fig). When analyzing inhibitory and excitatory neurons, nuclei showed signifi-cant differences in shape and texture features (S2C(i)–S2C(iv) Fig).

Having demonstrated the ability to distinguish between cell types using only nuclear fea-tures, we created a neural network model for classifying cell types based on their features (S4A Fig). Utilizing the dataset obtained from the 5 reporter mouse models, 5 neural network classi-fiers were trained, 1 for each cell type. The purpose of each classifier was to distinctly differen-tiate its corresponding cell type from the diverse array of other cell types (Figs 1C(i) and 3A). To increase the amount of training data and equalize the nuclei counts for each cell type, thus reducing training bias, synthetic data was generated from the features of the original dataset (S4C Fig). Synthetic data distribution fit well to the distribution of the original data for each cell type (S4C and S4D Fig). After each nucleus was classified by all trained classifiers, it was

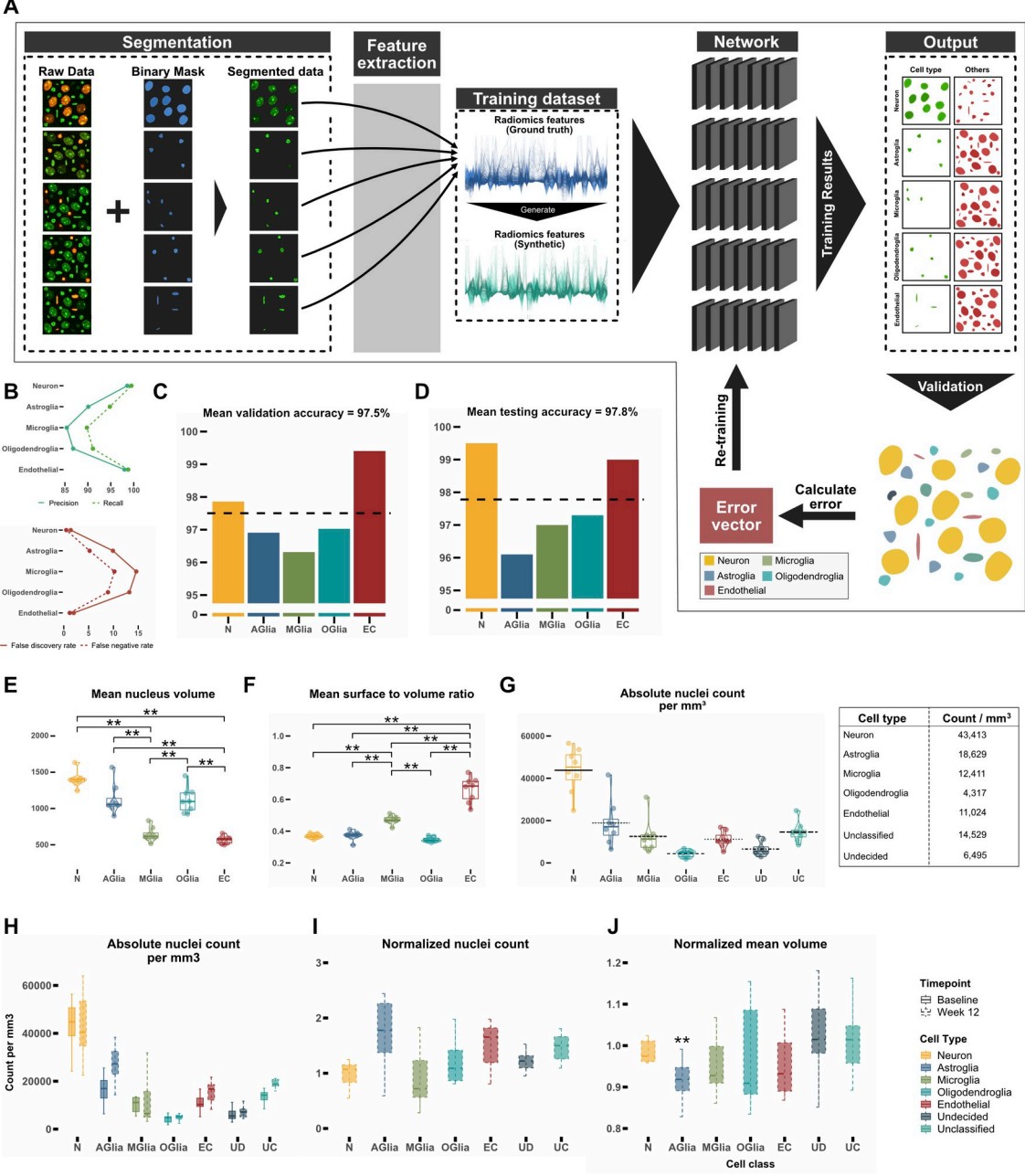

**Fig 3. Training a neural network for cell type classification.** **(A)** Schematic depiction of the training process. After segmenting nuclei of the raw data with StarDist for every cell type, radiomics features of nuclei were extracted from the ground truth data (blue) and synthetic data (green) was generated. For each cell type, a single neuronal network was trained to distinguish the corresponding nucleus type from the other nuclei. **(B)** Precision, recall, false discovery rate, and false negative rates for the test dataset. **(C, D)** Mean accuracy for all cell types in the validation dataset (15% of the ground truth dataset) and the test dataset (15% of the ground truth dataset). The dotted black horizontal line shows mean of all classifiers. **(E, F)** Comparison of mean nucleus volume (unit: µm$^3$) and mean nuclear surface to volume ratio (unit: µm$^{-1}$) of each cell type for classified volumes. **(G)** Number of nuclei per mm$^3$ in the secondary motor cortex. **(H)** Number of nuclei per mm$^3$ in the secondary motor cortex 12 weeks apart. Black horizontal line depicts mean per class. **(I)** Comparison of normalized number of nuclei after 12 weeks. **(J)** Mean nucleus volume after 12 weeks normalized to baseline. (Significance testing for C, D, E, F, G, H, I, J: Wilcoxon test, p-values were corrected for multiple comparisons using the Bonferroni method, n = 8 mice, $p < 0.05^*$, $p < 0.01^{**}$, $p < 0.001^{***}$), N = Neuron, AGlia = Astroglia, MGlia = Microglia, OGlia = Oligodendroglia, EC = Endothelial cells. Plot data can be found in S1 Data.

either assigned to a single class (neurons, astrocytes, microglia, oligodendroglia, and endothelial cells), 2 or more classes (undecided) or to none (unclassified). Precision and recall rates for the model were high for neurons and endothelial cells (Fig 3B). Due to their relative similarity, glial cells exhibited lower precision and recall rates. The classification accuracy for the entire training dataset was highest for neurons (98%) and endothelial cells (99%), whereas the classifier showed slightly lower accuracies for all glial cell types, especially for microglia (96%) (Fig 3C and 3D, for the respective confusion plots see S5 Fig). To be able to distinguish between inhibitory and excitatory neurons, another classifier was trained on the ground truth data from excitatory and inhibitory cells. The classifier had a 93% overall accuracy (S2D Fig), enabling a good distinction between inhibitory and excitatory neuronal cells. The 3D classification results for 1 example cortical volume containing 20,123 nuclei (S6A and S6B Fig) visualizes the relative density and positioning of cell types within the volume and their mutual spatial relationship (see also S1 Movie for an animated version of the segmentation and classification process).

To apply NuClear in mice, we studied 16 imaging volumes from 8 mice (male, 12 to 14 weeks at baseline) in layer 2/3 of the secondary motor cortex 12 weeks apart. When the trained model was applied to the first imaging time point (baseline), 2 features not chosen by the automated feature selection algorithm differed significantly between the classes: nucleus volume (Fig 3E) and mean surface to volume ratio, a measure for the sphericity of the nucleus (Fig 3F). Both findings are expected for each of the cell types and hence demonstrate that even features that were not used for training the neural network model are able to differentiate between the classes. At baseline, a density of 43,000 neurons per mm$^3$ and of 57,000 glial cells per mm$^3$ (including unclassified cells that were counted as glial cells) could be calculated, resulting in a glia to neuron ratio of 1.3 (Fig 3G). The density of excitatory neurons was 37,000 cells/mm$^3$ compared to 6,000 cells/mm$^3$ for inhibitory neurons (S2E Fig). The density of cells 12 weeks after baseline imaging remained unchanged after correcting for multiple comparisons (Fig 3H). When normalizing these counts to baseline, only the number of astrocytes shows a trend to increase (Fig 3I), which could be either due to astrogliosis caused by aging or by continued reaction to the chronic window implantation. Interestingly, astroglial nuclear volume significantly decreased over time (Fig 3J), which might be due to altered transcriptional activity known to occur in astrocytes with aging [12,13], whereas the nuclei volume in inhibitory and excitatory neurons did not change over time (S2F Fig). This latter result illustrates that our method can extract even subtle and unexpected changes from of a vast parameter space of different aspects of tissue composition.

## Discussion

Here, we present a novel method for intravital, longitudinal, comprehensive, large-scale, and 3D determination of tissue composition based on a deep learning algorithm trained on radiomics features of cell nuclei (NuCLear—Nucleus-instructed tissue composition using deep learning). We demonstrate that cell types and even subtypes can be reliably distinguished via the respective properties of their cell nucleus and that an inventory of cells and their 3D position can be generated for a large imaging volume. To demonstrate the usability of NuCLear, we analyzed volumetric images of the cortex of H2B-eGFP mice acquired with in vivo 2 photon microscopy. To be able to image a large dataset in a comparatively short time, we chose an imaging resolution with an axial resolution of 2 μm and a lateral resolution of 0.3 μm. We were able to image a whole 3D volume of 700 μm$^3$ consisting of approximately 25,000 cells in around 20 min. This speed will make it possible to perform large-scale data acquisition in a

repeated manner, enabling longitudinal intravital monitoring of tissue histology over time and in response to perturbations, treatments, or adaptive processes such as learning or exploring.

Neurons and endothelial cells showed the best classification results regarding total accuracy and precision as well as recall rates (Fig 3B), as their nucleus feature profile turned out to be markedly different between each other as well as between the individual glial cell types (Fig 2E and 2F). Differentiation between the 3 different glial cell types was more difficult to achieve, resulting in lower overall accuracy rates as well as lower precision and recall rates, due to the higher similarity of their respective nucleus feature profiles. Nevertheless, with an accuracy above 90% our approach can reliably be used to subclassify glia. We expect that a higher imaging resolution as well as more training data would further increase the classifier performance. Another reason for the comparatively lower classification accuracy in microglia might be due to acute injection of tomato-lectin to label these cells, which could have introduced a bias towards activated microglia. Nevertheless, the overall accuracy of the microglia classifier for the whole training dataset was around 96%, owing to the fact, that non-microglial nuclei could be identified correctly. For astrocytes, we used GFAP-labeled cells for classification, a label that primarily marks reactive astrocytes and has a higher expression in older animals. Future studies could optimize this labeling strategy. Additional confirmation of the classifier's precision can be inferred by examining characteristics of the nuclei, like average nuclear volume and the surface-to-volume ratio, which were not utilized in the training phase (Fig 3E and 3F). These features showed distinct clustering and low standard deviation in each class as well as significant differences between the cell types.

When applying the NuCLear to imaging volumes from layer 2/3 of the secondary motor area and calculating the number of cells per mm$^3$, a marked similarity of the number of neurons with published results is evident (43,413 ± 1,003 versus 45,374 ± 4,894) [1] (Fig 3G). Astrocyte numbers are similar (18,629 ± 1,007 versus 13,258 ± 1,416), microglia numbers differ more (12,411 ± 745 versus 24,584.6 ± 2,687). Oligodendrocyte numbers seem to be vastly different (4,317 ± 154 versus 51,436.1 ± 488), which might stem from the fact that only layer 2/3 was analyzed, excluding deeper cortical layers that have been shown to contain most oligodendrocytes [14]. When all glial cells as well as undecided and unclassified cells are summed up, a Glia/Neuron ratio of 1.3 can be calculated, a result that is in line with known results of the rodent cortex [15]. Candidates for unclassified cells could be the relatively large population of oligodendrocyte precursor cells (about 5% of all cells in the brain [16]). After classifying the neurons into subtypes, 86% of all neurons were classified as excitatory and 14% as inhibitory (37,199 ± 1,387 versus 6,214 ± 1,001, respectively), which is in par with previous studies that reported 10% to 20% of neurons in layer 2/3 of the cortex being inhibitory interneurons [17,18]. In conclusion, cell type counts are comparable to published results, supporting the validity of our approach.

A decisive advantage of NuCLear over existing ex vivo methods is the ability to study cell type changes over time in the same imaging volume and thus achieving a higher statistical power in experiments with fewer animals, which is crucial for complying with the "3R" rules in animal research (reduce, replace, refine). We succeeded to image selected locations up to 1 year after the implantation of chronic cranial windows. When analyzing the same imaging volumes from the secondary motor cortex 12 weeks apart, we were able to detect a trend towards astrogliosis, which might be due to the aging process leading to a higher GFAP-reactivity [19]. The significant decrease in mean nucleus volume of astroglia over time could be attributed to altered transcriptional activity associated with aging [12,13] since astroglial reactivity changes in the cortex over time [13]. Another factor underlying astrogliosis might arise from the continued presence of the chronic cranial window [20]. Our study illustrates only 1 possible application of NuCLear and subsequent analysis of 3D tissue composition. Here, we focused on the

identification of cell types, but the 3D datasets generated allow for additional analyses such as statistical distribution of cell types relative to each other, their nearest neighbor distances or inferences on physical tissue volume [21]. Our classifiers mostly rely on geometrical parameters such as nucleus shape, but also consider texture information to a certain degree. Pathophysiological conditions that alter nuclear shape and texture might influence classification accuracy. An increased sampling resolution may further increase textural information and thereby yield even higher classification accuracies. For now, segmentation and classification depend on the type of microscope used, quality of the images, and depth of imaging as well the region which was analyzed. To make the classification more robust and accountable for different qualities of fluorescence signal, augmentations to the training data could be added such as 3D blurring with PSF-shaped convolutions.

NuCLear will be applicable to different organs as techniques have emerged in the last years to perform in vivo imaging through chronic windows in other rodent organs such as skin, abdominal organs, the tongue, or spinal cord [22], depending on the image quality and the possibility to acquire images during multiple time points. Other organisms and in vitro preparations like cell cultures or organoids can also be utilized, provided that the respective ground truths can be generated. It will be usable with a variety of microscopy techniques such as confocal-, light-sheet-, or three-photon microscopy, making large-scale tissue analysis much more accessible. We assume that it will be possible to detect more cell types by adding classifiers trained with appropriate ground truth data. In this way, we propose a readily usable method to implement large-scale tissue analysis ex vivo or in vivo to study effects of interventions on cell type compositions of different organs.

## Materials and methods

### Ethical approval

The entire study was conducted in line with the European Communities Council Directive (86/609/EEC) to reduce animal pain and/or discomfort. All experiments followed the German animal welfare guidelines specified in the TierSchG. The study has been approved by the local animal care and use council (Regierungspräsidium Karlsruhe, Baden-Wuerttemberg) under the reference numbers G294/15 and G27/20. All experiments followed the 3R principle and complied with the ARRIVE guidelines [23].

### Animals

Adult transgenic mice expressing the human histone 2B protein (HIST1H2B) fused with eGFP under the control of the chicken beta actin promoter (CAG-H2B-eGFP) [24] (B6.Cg-Tg (HIST1H2BB/EGFP)1Pa/J, Jackson Laboratory; # 006069) were used in all animal experiments. For the reporter mouse lines, H2B-eGFP mice were bred to carry a floxed sequence of the red fluorescent protein tdTomato and a tamoxifen inducible Cre-recombinase under the expression of different cell type specific promoters: GFAP (Glial Fibrillary Acidic Protein) for astroglia (HIST1H2BB/EGFP-GFAP/ERT2CRE- CAG/loxP/STOP/loxP/tdTomato), PLP (Proteolipid Protein) for oligodendroglia (HIST1H2BB/EGFP-PLP/ERT2CRE- CAG/loxP/ STOP/loxP/tdTomato), and Tie2 (Tyrosine Kinase) for endothelial cells (HIST1H2BB/EGFP--Tie2/ERT2CRE- CAG/loxP/STOP/loxP/tdTomato). Expression of the Cre-recombinase was achieved over a course of up to 5 days with daily intraperitoneal injection of 2 doses of 1 mg Tamoxifen (Sigma Life sciences; dissolved in 1 part ethanol (99.8% absolute) and 10 parts sunflower oil) (Figs 2B and S1D). Microglia were visualized using intracortical injection of Lycopersicon Esculentum (Tomato) lectin coupled with DyLight 594 during the cranial window surgery after dilution by 1:39 in 150 mM NaCl, 2.5 mM KCl, 10 mM HEPES at pH 7.4 [25]

(Figs 2B and S1D). For neuronal labeling, an in house produced adeno-associated virus expressing mCherry under the Synapsin promotor was cortically injected during the cranial window surgery (Figs 2B and S1D). To label excitatory and inhibitory neurons, a viral labeling strategy was implemented via intracortical injections, using AAV5-CamKIIα-mCherry (Addgene plasmid 114469) and AAV1-mDLX-mRuby2 (Addgene plasmid 99130) (S2 Fig) [26]. All mice were between 10 and 18 weeks old during baseline imaging (5 female/10 male).

### Chronic cranial window implantation

A craniectomy procedure was performed on each mouse to enable in vivo two-photon imaging following a predefined protocol as described before [27] (S1A Fig). Briefly, the mice were anesthetized with an intraperitoneal injection (i.p.) of a combination of 60 μl Medetomidine (1 mg/ml), 160 μl Midazolam (5 mg/ml), and 40 μl Fentanyl (0.05 mg/ml) at a dosage of 3 μl/g body weight. The head was shaved, and the mice were fixed with ear bars in the stereotactic apparatus and eye ointment was applied. Xylocain 1% (100 μl, Lidocaine hydrochloride) was applied under the cranial skin and 250 μl Carprofen (0.5 mg/ml) was injected subcutaneous (s. c.). The skin was removed to expose the skull, and the surface of the skull was made rough to allow the cement to adhere better. A skull island (approx. 6 mm ∅) was drilled centered at 1 mm rostral to bregma using a dental drill and removed with a forceps (#2 FST by Dumont) making sure not to damage the cortical surface. For improved imaging condition, the dura was carefully removed from both hemispheres using a fine forceps (#5 FST by Dumont). Normal rat ringer solution (NRR) was applied on the exposed brain surface to keep it moist, and a curved cover glass was placed on top to cover it [21]. With dental acrylic cement (powder: Paladur, Kulzer; activator: Cyano FAST, Hager & Werken GmbH), the cover glass was sealed, and excess cement was used to cover the exposed skull and edges of the skin. A custom designed 3D printed holder was placed on top of the window and any gaps were filled with cement to ensure maximum adhesion with the skull. After the procedure, the mice were injected (i.p./s.c.) with a mix of 30 μl Atipamezole (5 mg/ml), 30 μl Flumazenil (0.1 mg/m), and 180 μl Naloxon (0.4 mg/ml) at a dosage of 6 μl per gram body weight. To ensure proper recovery of the mice, 3 more doses of Carprofen were given every 8 to 12 h, and the mice were placed onto a heating plate and monitored.

### In vivo two-photon imaging

Imaging was carried out with a two-photon microscope (TriM Scope II, LaVision BioTec GmbH) with a pulsed Titanium-Sapphire (Ti:Sa) laser (Chameleon Ultra 2; Coherent) at excitation wavelengths of 860 nm for DyLight 594 and 960 nm for H2GFP, tdTomato and mCherry, respectively. A water immersion objective (16×; NA 0.8, Nikon) was used to obtain volumetric 3D stacks. Individual frames consisted of 694 μm × 694 μm in XY with a resolution of 0.29 μm/pixel. Stacks were obtained at varying depths up to 702 μm from the cortical surface with a step size of 2 μm in Z. Prior to each imaging session, the laser power was adjusted to achieve the best signal to noise ratio. Adaptations were made to minimize the effect of laser attenuation due to tissue depth by creating a z-profile and setting the laser power for different imaging depths while making sure to minimize oversaturation of pixels.

Mice were initially anesthetized using a vaporizer with 6% isoflurane, eye ointment was applied, and mice fixed in a custom-built holder on the microscope stage. Isoflurane was adjusted between 0.5% and 1.5% depending on the breathing rate of each mouse to achieve a stable breathing rate of 55 to 65 breaths per minute with an oxygen flow rate of 0.9 to 1.1 l/ min. A heating pad was placed underneath the mouse to regulate body temperature. An infrared camera was used for monitoring of the mice during the imaging.

For ground truth training data reporter mouse lines for neurons and neuronal subtypes, astroglia, oligodendroglia, and endothelial cells were imaged 2 to 4 weeks after the cranial window surgery. Microglia reporter mice were imaged immediately after the cranial window surgery for a minimal inflammatory reaction (for a full list of reporter mice numbers, see S4G Fig). For the data obtained from adult H2B-eGFP mice used for classification, 3D volumetric stacks were imaged in the secondary motor cortex at 2 different time points. This included baseline imaging that was performed at 3 to 4 weeks after the chronic cranial window surgery and 12 weeks after the baseline time point. To investigate a possible age effect, mice underwent a sham surgery at the left hind paw 1 week after baseline surgery as part of a different study.

### Automatic nuclei segmentation using StarDist

Automated nuclei segmentation was achieved using the StarDist algorithm, which implements a neural network to detect star-convex polygons in 2D and 3D fluorescence imaging [10]. The StarDist model was trained to segment nuclei in 3D using an in-house developed Jupyter Notebook. A GUI based version of the software for training of the segmentation (NucleusAI) can be found here: https://github.com/adgpta/NuCLear. Nuclei were manually traced and segmented using the segmentation editor in Fiji [28] (S2 Movie). In total, 15 different crops from the two-photon volumetric data with approximately 150 to 200 nuclei each were used for training the StarDist 3D segmentation classifier (S1I Fig).

Detection accuracy and precision were calculated after visualization of raw data and segmentation in ImageJ/Fiji. True positive nuclei (TP), false positive nuclei (FP), and false negative nuclei (FN) were manually counted. Four arbitrary volumetric images were selected containing up to 80 nuclei per crop. Accuracy was calculated as TP/(TP + FP + FN). Precision was calculated as TP/(TP + FP).

### Feature extraction of segmented nuclei using PyRadiomics

Using the PyRadiomics python package [11], in total 107 radiomics features were extracted for each nucleus after segmentation using StarDist, including, but not limited, to morphological features, intensity features, and texture features (see S1 Table). Nuclei touching the borders of the imaging volume were excluded to avoid mistakes in the classification. Within an imaging volume of 700 μm³ containing approximately 25,000 nuclei, the number of border touching nuclei represented only 0.03% of the total number of nuclei, thus diminishing the possibility of any error caused by an edge effect.

### Extraction of data for training of the classifier

To perform supervised training of the deep neural network for cell type classification, a ground truth dataset was created using the two-photon volumetric fluorescence data, automatically segmented nuclei, which were assigned a unique label, and the radiomics features for each segmented nucleus. With the help of the red fluorescence channel images, nuclei belonging to a specific cell type were manually identified. Nuclei that were overlapping in the green and red channel images in the post-induction time point were reidentified in the green channel images of the preinduction time point. These nuclei were manually selected in the segmented StarDist output images. Using their label, corresponding radiomics features could be extracted from the preinduction time point image (S4B Fig). Approximately 70 to 400 nuclei were identified for each cell type (S4G Fig).

To increase the number of nuclei for training and make it equal in size for all cell types and thus avoid a bias in training, synthetic datasets were created from nuclei features using the synthetic data vault (SDV) package [29], which utilizes correlations between features of the

original dataset as well as mean, minimum or maximum values, and standard deviations. The synthetic data generated by the model fit to the original dataset with approximately 83% accuracy (S4D Fig).

## Training of the classification model

All training and validation of the classifiers were performed with a custom MATLAB (R2021a, Mathworks) script, visualization was performed in ImageJ/Fiji (S3 Movie). Original and synthesized data were merged into a single table and random indices were generated to divide the dataset into a training, test, and validation dataset with 70%, 15%, and 15% of the total dataset, respectively. Datasets with different amounts of synthetic data combined with the original dataset were created and training accuracy was later compared between them: orig = original dataset, orig * 2.5 = dataset containing 2.5 times the amount of data as the original dataset, orig * 2.5 (down sampled) = dataset down sampled to minimum sample count (after 2.5-fold increase) to equalize sample numbers for all cell types, orig * 9 = dataset containing 9 times the amount of data as the original dataset, orig * 9 (down sampled) = dataset down sampled to minimum sample count (after 9-fold increase) to equalize sample numbers for all cell types (S4E and S4F Fig). Down sampled datasets were created by selecting the class with the minimum number of nuclei and removing random samples from the other classes to match the nuclei count of the selected class. The same training, test, and validation datasets were used for training 5 different classifiers, 1 for each cell type. In the training dataset, for each classifier, the class of interest was assigned a unique identifier and all other cell classes were denoted by another identifier, for example, when training the neuronal classifier, the neurons were labeled as "Neuron" while all the other cell types in the dataset were labeled "Other" (Figs 1C(i), 3A and S4A). Each classifier had the same design. Initially, a sequential feature selection function "sequentialfs" (MATLAB R2022b) was applied to the whole dataset to extract 12 features with the highest variability to reduce data overfitting. These were fed into the "feature input layer" of the classifier. The features were z-score normalized for better compatibility between each class. A "fullyconnected" layer of 50 outputs was created and a mini-batch size of 16 samples was set to reduce the time and memory consumption during training. A batch normalization layer was added to reduce sensitivity to the initialization of the network. The activation layer ReLU (rectified linear unit) was added for thresholding operations. Another "fullyconnected" layer was added, which sets the number of outputs identical to the number of input classes, and a second activation layer "softmax" was chosen for classifying the output into the 2 separate groups. The "adam" adaptive moment estimation optimizer [30] was selected as the solver for the training network that updates the network weights based on the training data in an iterative manner. The training data and the validation data were shuffled every epoch, i.e., a complete pass or iteration through the entire training dataset during the training process, before training and network validation, respectively. After each classifier was trained, further automatic and visual validation was performed to check for accuracy.

## Classification of the H2B-eGFP data

After acquiring volumetric images of H2B-eGFP nuclei ($n = 8$ mice, male), automated segmentation of nuclei was performed with the StarDist neuronal network (see above, S6A and S6C Fig). Features for the segmented nuclei were extracted using PyRadiomics (see above) and then classified with all the different classifiers trained on the orig * 9 dataset (see "Training of the classification model" for a description of the dataset). Thus, a single nucleus would be either labeled as belonging to one of the 5 classes or to the "Other" class. Nuclei that were assigned to multiple classes were labeled as "Undecided" (UD). Any nuclei that were identified

as "Other" by all the classifiers were labeled as "Unclassified" (UC). Nuclei of neurons were further subdivided into excitatory and inhibitory subtypes.

## Statistical analysis

All classified nuclei and their features were stored in a local MySQL database (MySQL Workbench 8.0). The data were imported into R [31] for statistical analysis. Cell count, mean nuclei volume, and first nearest neighbor distance were calculated for each cell type for different time points. Since the distribution of the data was nonparametric, the Wilcoxon signed-rank test was used for all statistical testing and $p$-values were corrected for multiple comparisons using the "bonferroni"-method. Plots were created using the R package "ggplot2" [32].

## Supporting information

**S1 Fig.** **(A)** **(i–iii)** Chronic cranial window implantation using a curved glass cover slip and a custom 3D printed holder. **(iv)** Fixation of the anesthetized mouse in the custom holder for imaging under the two-photon microscope. **(B)** X-Z maximum intensity projection of a two-photon volumetric stack (700 μm × 700 μm × 700 μm). **(C)** Imaging position on a DAPI-stained brain slice. Slice thickness 50 μm. DAPI intensity profile showing similar distribution of intensity as a two-photon image of nuclei. The decreasing nucleus density in the two-photon image stack is a result of attenuation of fluorescence signal at higher depths. Inlay: Nucleus density distribution of sub-volumes used for data analysis. **(D)** Scheme showing intracranial and intraperitoneal injection strategies for labeling cell types. **(E)** Fluorescence emission spectra for eGFP, tdTomato, and mCherry (y axis: fluorophore emission normalized to quantum yield, source: fpbase.org). **(F)** Labeling strategies for red fluorescence expression in reporter mouse lines. Visualization of crosstalk between the eGFP and tdTomato signal after induction with tamoxifen. Overlay of pre (cyan) and post (yellow) GFP after image alignment with the ImageJ plugin bUnwarpJ [33]. **(G)** No crosstalk is visible between GFP and mCherry signals (upper panel) or eGFP and Tomato lectin-Dylight 594 signals (lower panel). **(H)** (i) Raw data of H2B-eGFP signal, (ii) manually labeled ground truth, (iii) StarDist segmentation, (iv) composite image of ground truth (red), and StarDist segmentation (green). **(I)** Upper panel: Count of manually segmented ground truth nuclei for StarDist training, each color depicts an individual mouse. Lower panel: StarDist nucleus detection accuracy and precision, each bar represents an individual imaging volume. Plot data can be found in S1 Data.
(TIFF)

**S2 Fig. Training a classifier to distinguish excitatory from inhibitory neurons. (A)** Labeling strategies for excitatory and inhibitory neurons. Intracortical injections were performed at approximately 400 μm deep from the cortex of H2B-eGFP mice using AAV5-CamKIIα-mCherry and AAV1-mDLX-NLS-mRuby2 to visualize excitatory and inhibitory neurons. **(B)** Post-injection images. Inhibitory neurons were imaged 2–3 weeks after injection; excitatory neurons were imaged 3–4 weeks after injection. **(C)** Radiomics features showing differences between excitatory and inhibitory neurons for (i) surface area (in pixel$^2$ [resolution x = 0.29 μm, y = 0.29 μm, z = 2 μm]), (ii) maximum 2D diameter (in pixel, same resolution as in (i)), (iii) first order 10th percentile, (iv) mesh volume in voxels (resolution as in (i)), ($n$ of excitatory neurons: 396, $n$ = 3 mice, $n$ of inhibitory neurons = 122, $n$ = 2 mice). **(D)** Confusion plot for the classifier. Rows show the predicted class (output class), and the columns show the true class (target class). Green fields illustrate correct identification whereas blue fields illustrate erroneous identifications. The number of observations and the percentage of

observations compared to the total number of observations are shown in each cell. Column on the far right shows the precision (or positive predictive value) and false discovery rate in green and red, respectively. Bottom row denotes recall (or true positive rate) and false negative rate in green and red. Cell on the bottom right shows overall accuracy of the classifier. **(E)** Number of nuclei per mm$^3$ in the secondary motor cortex at baseline after 12 weeks (dashed line) ($n = 8$ mice). **(F)** Mean nucleus volume after 12 weeks normalized to baseline ($n = 8$ mice). (Significance testing for C, E, F: Wilcoxon test, $p$-values were corrected for multiple comparisons using the Bonferroni method, $p < 0.05^*$, $p < 0.01$ **, $p < 0.001$ ***) Plot data can be found in S1 Data.
(TIFF)

**S3 Fig. Combinations of nuclear morphology, intensity, and texture features show clear distinction between cell types. (A)** Flatness, gray level non-uniformity. **(B)** Minor axis length, flatness. **(C)** First order minimum, elongation. **(D)** Large area emphasis, first order 10th percentile. **(E, F)** Combinations of features in 3 dimensions. Plot data can be found in S1 Data.
(TIFF)

**S4 Fig. (A)** Visualization of the entire classification training process. After ground truth data were selected, a sequential forward feature selection algorithm was applied to extracted features from all nuclei of all cell types, which selected 12 features from the 107 radiomics features. Synthetic data was generated from all the radiomics features of all nuclei and all cell types. The combined dataset was used to train a classifier with the training, validation, and test data comprising 70%, 15%, and 15% of the combined dataset. **(B)** Example showing manual selection of ground truth data for supervised training. Post-induction green and red channels were overlayed to create a composite. Any nuclei appearing yellow (possessing green and red fluorescence) were identified in the preinduction GFP channel and from its corresponding segmentation, a label id was acquired and later used to identify the extracted features. **(C)** Synthetic training data generated from the original dataset matched the features of the original datasets. **(D)** Statistical analysis of similarity between the distribution of original data and distribution of synthetic data (K-S test; mean = 83.68%). **(E)** Datasets with different amounts of synthetic data were created and training accuracy was compared between them, orig = original dataset, orig * 2.5 = dataset containing 2.5 times the amount of data as the original dataset, orig * 2.5 (down sampled) = dataset down sampled to minimum sample count (after 2.5-fold increase) to equalize sample numbers for all cell types, orig * 9 = dataset containing 9 times the amount of data as the original dataset, orig * 9 (down sampled) = dataset down sampled to minimum sample count (after 9-fold increase) to equalize sample numbers for all cell types. **(F)** Mean accuracy of classifiers trained 5 times with different combinations of synthetic data; error bars denote standard deviation (SD). Light blue = orig * 2.5, dark blue = orig * 2.5 (downsampled), green = orig * 9, yellow = orig * 9 (down sampled), red = orig. **(G)** Table listing the amount of manually identified nuclei used in the training process (Excitatory N: excitatory neurons, Inhibitory N: inhibitory neurons). Plot data can be found in S1 Data.
(TIFF)

**S5 Fig. Confusion matrices for each classifier for the test dataset.** Since each classifier distinguishes between the desired class and every "other" class, the confusion matrix consists only of 4 fields. Rows show the predicted class (Output Class), and the columns show the true class (Target Class). Green fields illustrate correct identification of target and "other" class, blue fields illustrate erroneous identifications. The number of observations and the percentage of observations compared to the total number of observations are shown in each cell. Column on the far right shows the precision (or positive predictive value) and false discovery rate in green

and red, respectively. Bottom row denotes recall (or true positive rate) and false negative rate in green and red. Cell on the bottom right shows overall accuracy of the classifier. Plot data can be found in S1 Data.
(TIFF)

**S6 Fig.** **(A)** Three-dimensional representation of raw, segmented, and classified data (700 μm × 700 μm × 700 μm volume) visualizing comprehensive cell type composition. **(B)** Cell type distribution in a single volumetric stack. **(C)** Maximum intensity z-projection of a sub volume (150 μm × 150 μm × 100 μm) showing raw, segmented, and classified nuclei in the X-Y axis.
(TIFF)

**S1 Table. Radiomics features extracted from segmented nuclei.** Features marked in green were used for cell type training and classification.
(TIF)

**S2 Table. Morphology, intensity, and texture features used for training and classification.**
(TIF)

**S1 Text. Explanations for technical terms used in the manuscript.**
(DOCX)

**S1 Movie. Illustration of nucleus segmentation and classification using NuCLear.** Raw data of H2B-eGFP-labeled nuclei (green), nucleus segmentation (randomly colored), and classification (apricot = neurons, red = microglia, blue = oligodendrocytes, dark green = astrocytes, magenta = endothelial cells).
(MP4)

**S2 Movie. Visualization of the manual nucleus segmentation process.**
(MP4)

**S3 Movie. Visualization of a classified 3D volume.**
(MP4)

**S1 Data. The excel sheet contains all data that were used to generate Figs 2F, 3C–3J, and S1I, S2C–S2F, S3A–S3F, S4D, S4F and S5.** Data can be found on sheets within the excel document named after the corresponding figure and panel.
(XLSX)

## Acknowledgments

We thank Michaela Kaiser for her excellent technical support during preparation and execution of experiments. We thank Sidney Cambridge for devising breeding schemes for mouse lines needed for ground truth generation and for help with mouse breeding.

## Author Contributions

**Conceptualization:** Johannes Knabbe.

**Data curation:** Amrita Das Gupta, Livia Asan, Jennifer John, Johannes Knabbe.

**Formal analysis:** Amrita Das Gupta, Livia Asan, Johannes Knabbe.

**Investigation:** Amrita Das Gupta, Livia Asan, Johannes Knabbe.

**Methodology:** Amrita Das Gupta, Johannes Knabbe.

**Project administration:** Thomas Kuner, Johannes Knabbe.

**Resources:** Thomas Kuner.

**Software:** Amrita Das Gupta, Carlo Beretta.

**Supervision:** Thomas Kuner, Johannes Knabbe.

**Visualization:** Amrita Das Gupta.

**Writing – original draft:** Amrita Das Gupta, Johannes Knabbe.

**Writing – review & editing:** Thomas Kuner, Johannes Knabbe.

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
