## [Editor Report · Decision Letter 0]

6 Feb 2023

Dear Dr Knabbe, 

Thank you for submitting your manuscript entitled "Comprehensive monitoring of tissue composition using in vivo imaging of cell nuclei and deep learning" for consideration as a Methods and Resources Article by PLOS Biology. Please accept my sincere apologies for the delay in getting back to you as we consulted with an academic editor about your submission. 

Your manuscript has now been evaluated by the PLOS Biology editorial staff, as well as by an academic editor with relevant expertise, and I am writing to let you know that we would like to send your submission out for external peer review.

Once your full submission is complete, your paper will undergo a series of checks in preparation for peer review. After your manuscript has passed the checks it will be sent out for review. To provide the metadata for your submission, please Login to Editorial Manager (https://www.editorialmanager.com/pbiology) within two working days, i.e. by Feb 08 2023 11:59PM.

Kind regards,

Richard

Richard Hodge, PhD

Associate Editor, PLOS Biology

rhodge@plos.org

PLOS

---

## [Decision Letter · Decision Letter 1]

21 Mar 2023

Dear Dr Knabbe,

Thank you for your patience while your manuscript "Comprehensive monitoring of tissue composition using in vivo imaging of cell nuclei and deep learning" was peer-reviewed at PLOS Biology. Please accept my apologies for the long delays that you have experienced during the peer review process. It has now been evaluated by the PLOS Biology editors, an Academic Editor with relevant expertise, and by two independent reviewers. 

In light of the reviews, which you will find at the end of this email, we would like to invite you to revise the work to thoroughly address the reviewers' reports.

As you will see below, the reviewers find think you approach is interesting and potentially useful, but raise overlapping concerns with the accessibility and reproducibility of the method since the code has not yet been made publicly available and that the scripts/methods should be assembled into an easily accessible code package that can be used as a resource by the community. In addition, the reviewers note that the utility of the method should be broadened to include additional datasets that classify neurons into subtypes, as demonstrated with glial cells. Please note that these concerns would need to be addressed in order for us to consider a revised version of the manuscript. 

Given the extent of revision needed, we cannot make a decision about publication until we have seen the revised manuscript and your response to the reviewers' comments. Your revised manuscript is likely to be sent for further evaluation by all or a subset of the reviewers.

**IMPORTANT - SUBMITTING YOUR REVISION**

*Re-submission Checklist*

*Published Peer Review*

*PLOS Data Policy*

*Blot and Gel Data Policy*

Sincerely,

Richard

Richard Hodge, PhD

Associate Editor, PLOS Biology

rhodge@plos.org

REVIEWS:

Reviewer #1: In this paper, Das Gupta et al describe an interesting approach for labelling major cell types in the brain using a computational method. I think the approach is clever, and could potentially be useful for many researchers. However, this is a computational method paper without code, which I can never recommend for publication. The authors must share their code as an easily accessible package, including clear instructions, scripts with comments, example applications etc. I would ask that they share the code with the reviewers for peer review purposes, and then commit to releasing this code publicly at journal publication. Similarly, the data should be made available publicly at publication date (it currently is not). The way the code is described in the methods makes me think there will be some work needed to put all the scripts and methods together into an easily accessible code package, and this is the work that I as a reviewer would like to request. I will make other comments below, which are of a secondary nature and much less important. 

Other comments: 

I think it should be stated very explicitly that this method can only categorize shallow cell types in the brain, i.e. neurons, astroglia, microglia etc. For the neuroscience community, cell types in the brain usually refers to types of neurons, such as "chandelier cells", or "PV-positive" cells, and these cannot be classified using this approach. 

Another thing that should be stated explicitly is that the brain is the only body region where a clear coverslip can even be implanted for in vivo imaging. This is very difficult to do for any other body organ, and the data quality from those implants is much much worse, precluding the application of these computational methods which need fine visual features to work. 

Another caveat of the method that should be stated is that it will depend on the type of microscope, depth of imaging and overall quality of the preparation. This makes a classifier trained on one set of data less robust at classifying new types of data, and it could explain the conflicting results which the authors report in the discussion with respect to the motor cortex volume imaged. To improve the ability of the decoder to generalize, the authors or other users could add augmentations to the training data to make it more similar to their own data, such as 3D blurring with PSF-shaped convolutions. This doesn't need to be done here, but I think it is important to give potential users options and ideas for what to do if the method does not work out of the box. 

Reviewer #2: The authors have presented a novel method for automatic classification of 5 major cell types of cortical layer 2/3: neurons, astroglia, microglia, oligodendroglia and endothelial cells. The method requires two neural networks, one for nuclei segmentation, the other for cell-type classification based on features of the nuclei. Both neural networks require training via ground-truth datasets, where the ground-truth datasets are created manually. After the initial training, the neural networks can be used in longitudinal studies where a given block of tissue is imaged over an extended period time. Hence, the NuCLear method presented by the authors could prove useful for researchers studying tissue composition in the brain and perhaps other types of tissue. However, it is not clear what resources the authors intend to provide besides the information contained in the manuscript; NuCLear is not a software package but a two-step neural-network method for classifying major cell types.

Major Comments:

1. The abstract is short and heavily weighted to "selling" the authors' NuCLear method with overreaching claims and very little mention of their experimental results (one brief sentence, line 22-23). The authors have not included the type of animal used in the study, what part of the brain that was investigated nor what major cell types were classified.

2. The order in which the authors present their Results is confusing. This is not helped by everything being presented in two long paragraphs. It appears the authors have started their Results with an overall summary with 3 main points: volumetric imaging of nuclei (GFP), neuronal network training using GFP nuclei and ground-truth data created via tdTomato and mCherry, classification of 5 main cell types. Perhaps the authors can make it clear they are summarizing their method from beginning to end here, and the remaining Results will be about filling in the details. However, the authors have not included their method of nuclei segmentation in their initial summary. Finally, the section on "Automated segmentation of nuclei in the raw data" (line 91) comes after the section describing the generation of ground-truth datasets via tdTomato and mCherry (line 73). This may be confusing since generation of the ground-truth datasets requires segmentation of the nuclei. I would suggest the authors present their Results in the same order as their Methods.

3. There is little information content in Figure 1. Moreover, the stripped-down simplicity of the cartoon seems to imply the NuCLear method is easy to use and requires few methodological steps. However, after looking at Figure 3a and Supplementary Figure 3, and reading the Methods, one begins to realise the numerous tasks necessary to perform the NuCLear method. There are actually two neural networks that are required, one for segmentation of the nuclei and the other for cell-type classification. Both neural networks require training via ground-truth datasets. Moreover, the ground-truth datasets have to be computed manually. I think Figure 1 should be replaced with Figure 3a or some version of it; it could also include information in Supplementary Figure 3. Or a simple flow box diagram could be useful.

4. The authors' figures are inadequately labelled making it frustratingly difficult to read/review their manuscript. The panel labels in the figures (a, b, c…) often do not match the labels in their legends (the authors have apparently rearranged their figures without rearranging the legends). Several graphs are missing units. There are many panels that are not adequately described in the legends.

5. The authors may not have computed their densities correctly. On line 317 they state: "Nuclei touching the borders of the imaging volume were excluded." To compute density correctly they need to consider the border effect (Gundersen 1977). For a volume, this means 3 walls of their volume should be "exclusive" and the other 3 "inclusive" when counting the nuclei. Gundersen HJG. Notes on the estimation of the numerical density of arbitrary profiles: the edge effect. J Microsc. 1977 Nov;111(2): 219-223. doi: 10.1111/j.1365-2818.1977.tb00062.x

6. From what I can tell from the Results the authors have only applied their method to one 3D volume of the cortex (line 129-130: "The 3D classification results for one example volume containing 20123 cells are shown in Figure 3E"). I would expect that after the authors trained their neural networks they would then move on to classifying neurons/glia in tissue that was not used in the training process, in multiple volumes as a proof of principle.

7. The authors have not discussed/explained why they did not attempt to classify neurons into subtypes as they did with glia. I would think the NuCLear method would find wider application if it was possible to categorize neuronal subtypes.

8. The authors have not clarified how much training of the neural networks will be necessary to perform their NuCLear classification method. Will the training only be necessary at the beginning of a study and then be valid thereafter for weeks/months/years on multiple preparations? Could the neural networks be shared between labs? Or will it be necessary for each lab to train their own neural networks?

9. There is a lack of experimental quantification in methods, e.g. the number, age and type (male/female) of mice, the number of imaged volumes used in the various steps of the NuCLear method.

10. It is not clear what data/code/networks (i.e. Resources) the authors plan to make available for others to use. Will a lab wanting to perform the NuCLear method have to replicate the authors' study from scratch?

11. There are several terms that are never defined in the manuscript: shape segmentation, binary mask, flatness, grey level non-uniformity, entropy, first order minimum, coarseness, overfitting, pixel entropy, confusion plots, radiomics, crops, etc.

Specific (minor) Comments

Title:

I think "monitoring" is not the best word for the title. The paper is more about classification of cell types than about monitoring cells over time.

Abstract:

Line 18. "Comprehensive analysis of tissue composition has so far been limited to ex-vivo approaches." This sentence is very general in scope and is not correct. I can think of MRI studies as one example.

Line 21. "This allowed us to classify all cells per imaging volume". This statement is not correct. There was a large number of unclassified and undecided cells in this study (Figure 3). And segmentation was not perfect.

Line 23. "…using only a single label." This statement is not correct. Besides labelling the nuclei with GFP, the authors labelled individual cell types with tdTomato and mCherry.

Line 24. "NuCLear opens a window to study changes in relative abundance and location of all major brain cell types…" The phrase "all major brain cell types" may be misleading. On first reading I was anticipating the authors meant that their method would be used to classify major neuronal types (pyramidal, stellate, etc.) but in fact neurons are all grouped into one category.

Line 25. "…enabling comprehensive studies of changes in cellular composition in physiological and pathophysiological conditions". If the pathology changes the shape of the nuclei then perhaps the authors' method will no longer work. The phrase "cellular composition" usually refers to intracellular composition, not cell-type composition of a tissue.

Line 26. "NuCLear will work with any fluorescence-based microscopy and perform in any organ or model system." The authors should be more careful with their choice of words such as "all" and "any" and "every". Other fluorescence-based microscopy methods might not have the necessary resolution to allow segmentation and deep learning in 3D. Not all organs, or parts of the brain, can be accessed using a chronic window implant.

Introduction:

Line 35: "all brain cell types" would be better as "multiple cell types".

Line 35-38: "…each cell type would require an individual labelling strategy using a set of specific promoters and fluorophores. This would require prioritizing cell types from the outset and thereby limit the scope to a small subset of the population". Are the authors implying their method resolves this problem? If that is so, then I am not sure how since their method has a labelling strategy using promoters and fluorophores.

Line 46: "all brain cell types". See comment above (line 35).

Line 49-50: "in the same mouse" is probably not what the authors are implying here.

Line 50: "…using only one genetically modified mouse line" should be "using a single genetically modified mouse line".

Line 51: "Histon" should be "histone". This typo occurs elsewhere.

Line 54-55: "In addition, knowing the 3D location of every cell enables previously inaccessible analyses." This sentence is vague and should be reworded to be more specific. 

Line 56-57: "…the concept of this technique will be applicable to many other cellular imaging techniques and in different experimental systems…" This phrase is vague and should be reworded to be more specific.

Results:

Line 67: "…where known cell types were additionally labelled with a red fluorescent protein (tdTomato or mCherry)". It is not clear what "additionally labelled" means. The green label should be mentioned here.

Line 70: "The trained neuronal network was used to classify cell types based on their GFP-labeled nuclei." This sentence seems redundant with the previous sentence.

Line 71-73: "In our study we selected five different cell types for classification Neurons (yellow), astroglia (blue), microglia (green), oligodendroglia (turquoise), and endothelial cells (red)". A colon is missing after "classification". Colours would be better listed in the figure legend.

Line 73-76: "To generate ground-truth datasets without a bleed through of the strong tdTomato fluorescence towards the GFP-channel (Supplementary Figure 1F), we chose a method of inducing the expression of tdTomato for unequivocal cell type identification after imaging of the nuclei." I suggest starting a new paragraph here. Why is bleed-through a problem for tdTomata and not mCheery? The word "unequivocal" is not correct since type identification is not perfect.

Line 76-78: "We selected a labelling strategy consisting of a tamoxifen-inducible cell type-specific Cre-recombinase in conjunction with…" The information in this sentence seems better placed before the previous sentence. A new paragraph could start: "To generate ground-truth datasets, we selected a labelling strategy consisting of…"

Line 79-80: "We acquired volumetric images of nuclei from these mouse lines and induced tdTomato expression via intraperitoneal injection of tamoxifen." The authors could move their tdTomata bleed-through information to this sentence.

Line 81: "the same location" is singular, but in previous sentence volumetric imaging is plural.

Line 81-83: "the previously imaged nuclei were selected by overlay with the red fluorescence signal." Readers may not understand this phrase. The authors should mention the green signal and those nuclei with both green and red overlap are selected. Was selection manual or automated?

Line 83-84: "As microglia changed their positions after induction with tamoxifen…" Was this effect only for microglia? If so, the authors should clarify this.

Line 88-90: "Hence, this approach was used to ascertain images of the labeled nuclei to be undisturbed by signals possibly arising from the red fluorescence used for unequivocal identification of the cell type." This sentence is confusing. Is it a summary sentence for all cell types or just neurons? As before, the word "unequivocal" is incorrect.

Line 91: "Automated segmentation of nuclei in the raw data…" This is the first mention of segmentation and it is not sufficiently described here. At what stage(s) does segmentation occur during the NuCLear method?

Line 106: "Having demonstrated the ability of nuclear features to distinguish between their corresponding cell types…" The authors should begin a new paragraph here.

Line 108-100: "Using data from the five reporter mouse lines, we trained one classifier per cell type for distinguishing one cell type from the respective other cell types." This sentence is difficult to understand and should be reworded.

Line 110-111: "To increase the amount of training data and equalize the nucleus counts for each cell type to reduce training bias…" This phrase has run-on "to" prepositions.

Line 114-115: "After classifying the whole dataset, nuclei were either assigned to a single class, two or more classes (undecided) or to none (unclassified)." The phrase "After classifying the whole dataset" is confusing: this sentence is about classifying. The authors should remind the reader what their different classes are.

Line 116-117: "Precision and recall rates for each classifier were high for neurons and endothelial cells." "Precision" and "recall" are not defined. Reference to Figure 3 in next sentence should be moved to this sentence.

Line 117-118: "After combining all classifier output…" It is not clear what this means.

Line 118: "Accuracy" is not defined. Readers may not understand the difference between precision and accuracy.

Line 129-130: "The 3D classification results for one example volume containing 20123 cells…" The authors should start a new paragraph here.

Line 130: I do not find Figure 3E and supplementary Figure 6 "intuitive".

Discussion:

Line 150: "all cells" is not correct.

Line 153: "feasible" is not the correct word to use here.

Line 157: "perturbations, treatments, or adaptive processes." These could be clarified, especially "adaptive processes".

Line 158-159: "…the best classification results regarding total classification accuracy and precision…" The second "classification" is redundant.

Line 171: "expressed increasingly" should be reworded.

Line 173: "concluded" is not the correct word to use here.

Line 177-190: "When applying the classification model to a Layer 2/3 volume…" This paragraph could be incorporated into the authors' Results.

Line 178: "…extrapolating the number of cells per mm³…" It is not clear what this means.

Line 196-197: "When analyzing the effect of time on the imaging volumes from the secondary motor cortex…" I could not find the "time" aspect in the Results or figures.

Line 204-205: "Our study illustrates only one possible application of NuCLear and subsequent analysis of 3D issue composition." The authors have not presented an application of their method, only details of the method itself.

Line 211: The word "easily" is not correct.

Line 214-215: "We anticipate that any cell type will be detected…" This is a bold claim.

Line 216: "…we propose an easy and readily usable method…" It does not seem to me the authors' method is easy nor readily accessible. It requires someone trained in neural networks.

Line 217: The authors have mentioned the use of their method for "different organs" in multiple places but have not clarified what organs they are referring to. Most organs are not accessible for in vivo imaging.

Materials and Methods:

Line 275: "Two photon imaging was carried out with a two-photon microscope". "Two photon" is redundant.

Line 303-304: "The StarDist model was trained to segment nuclei in 3D using an in-house developed Jupyter Notebook". There does not seem to be enough information here for other labs to develop their own in-house segmentation model.

Line 304: "Manually segmented and labelled nuclei…" It was not explained how nuclei were manually segmented/labelled. This may be non-trivial.

Line 313: "Feature extraction using PyRadiomics". The authors should add "nuclei" to the header.

Line 315: "…for each segmented nucleus…" The authors do not explain which nuclei dataset they are using here.

Line 320: "For supervised training of the deep neural network for cell type classification…" This phrase has run-on "for" prepositions.

Line 324: "label ID" is not defined.

Line 327-328: "To increase the number of nuclei for training and make it equal in size for all cell types to avoid a bias in training…" This phrase has run-on "to" prepositions.

Line 336: "manual validation was performed in ImageJ/Fiji". The method for this was not explained.

Line 353: Do the authors have a reference for the "adam" adaptive moment estimation optimizer?

Line 356: It is not clear what "epoch" refers to.

Line 361: "all nuclei" is not correct.

Line 363: "every segmented nucleus" should be "the segmented nuclei".

Line 364: "Every segmented nucleus" should be "The segmented nuclei".

Line 360-367: This section is confusing as it mostly repeats the previous methods.

Line 374: the authors have not specified what type of t-test they used.

Figures:

Figure 1: "all cell nuclei" is not correct. Missing information: mouse, type of mouse, type of fluorescence. There is no "c" label in the figure. Cell types and colours are not defined. It looks like the endothelial cells line the blood vessels but this is not explained by the authors.

Figure 2: Labels in legend do not match those in the figure. There are no "3D renderings" in the figure. Diameters have no units. It is not clear what "algorithm" the authors are referring to in the legend. The results in this figure may be confusing since the nuclei have been separated into cell types (panels e and f) before the authors have explained how they classified their neurons via neural networks (Figure 3).

Figure 3a: It is not clear what is plotted under "radiomics features" and "neural network classifier". There are no units for volumes or surface-to-volume ratios.

Supplementary Figure 3: "A sequential forward feature selection algorithm was applied." I am not sure if this algorithm was mentioned in the manuscript.

---

## [Decision Letter · Decision Letter 2]

6 Sep 2023

Dear Dr Knabbe,

Thank you for your patience while we considered your revised manuscript "Comprehensive monitoring of tissue composition using in vivo imaging of cell nuclei and deep learning" for publication as a Methods and Resources Article at PLOS Biology. This revised version of your manuscript has been evaluated by the PLOS Biology editors, the Academic Editor and the original reviewers.

Based on the reviews, I am pleased to say that we are likely to accept this manuscript for publication, provided you satisfactorily address the remaining points raised by Reviewer #2. Please also make sure to address the following data and other policy-related requests that I have provided below (A-G):

(A) We would like to suggest the following modification to the title: 

“NuClear enables accurate classification of major brain cell types during in vivo imaging using neural network processing”

(B) In the Financial disclosure statement in the online submission form, please provide any funding sources that you received to conduct the study, along with the corresponding grant numbers. 

(C) You may be aware of the PLOS Data Policy, which requires that all data be made available without restriction: http://journals.plos.org/plosbiology/s/data-availability. For more information, please also see this editorial: http://dx.doi.org/10.1371/journal.pbio.1001797

-Supplementary files (e.g., excel). Please ensure that all data files are uploaded as 'Supporting Information' and are invariably referred to (in the manuscript, figure legends, and the Description field when uploading your files) using the following format verbatim: S1 Data, S2 Data, etc. Multiple panels of a single or even several figures can be included as multiple sheets in one excel file that is saved using exactly the following convention: S1_Data.xlsx (using an underscore).

-Deposition in a publicly available repository. Please also provide the accession code/DOI so that we may view your data before publication. 

Figure 2F, 3C-J, S1I, S2C-F, S3A-F, S4D, S4F, S5

(D) Thank you for already depositing the code and examples in Github (https://github.com/adgpta/NuCLear). At this stage, we ask that you please attach this deposition to the Zenodo data repository to ensure long term maintenance and that the deposition is assigned a DOI. Please ensure that the code is sufficiently well documented and reusable, and that your Data Statement in the Editorial Manager submission system accurately describes where your code can be found.

(E) Please also ensure that each of the relevant figure legends in your manuscript include information on *WHERE THE UNDERLYING DATA CAN BE FOUND*, and ensure your supplemental data file/s has a legend.

(F) Please ensure that your Data Statement in the submission system accurately describes where your data can be found and is in final format, as it will be published as written there. 

(G) Please also provide a blurb which (if accepted) will be included in our weekly and monthly Electronic Table of Contents, sent out to readers of PLOS Biology, and may be used to promote your article in social media. The blurb should be about 30-40 words long and is subject to editorial changes. It should, without exaggeration, entice people to read your manuscript. It should not be redundant with the title and should not contain acronyms or abbreviations. For examples, view our author guidelines: https://journals.plos.org/plosbiology/s/revising-your-manuscript#loc-blurb

We expect to receive your revised manuscript within two weeks. 

*Published Peer Review History*

*Press*

Sincerely,

Richard

Richard Hodge, PhD

rhodge@plos.org

Reviewer remarks:

Reviewer #1: The authors have addressed my concerns. I do not have any follow-up concerns.

Reviewer #2 (Jason Seth Rothman, signs review): "Comprehensive monitoring of tissue composition using in vivo imaging of cell nuclei and deep learning"

The latest version of the authors' manuscript has improved considerably from their initial version. The authors have addressed all of my major and minor concerns. I have only a few remaining minor concerns about the revised manuscript.

Line 18. "Comprehensive analysis of tissue composition has so far been limited to ex-vivo approaches." The authors state they changed this sentence but it has not changed. My example of MRI was to highlight how vague this sentence is, which does not even include the word 'imaging'.

Lines 26 and 147: 'relative abundance' would be better as 'relative density'.

Line 28: 'cellular composition' would be better as 'tissue composition' or 'cell-type composition' as used elsewhere in the manuscript.

Line 32-34: 'Understanding the plasticity and interaction of different brain cell types in vivo has been a limiting factor for the investigation of large-scale structural brain alterations associated with diverse physiological and pathophysiological states.' This sentence is confusing. It is not clear what the 'limiting factor' is.

Line 39-40: 'studies assessing whole tissue composition quantitatively' would be better as 'quantitative studies assessing whole tissue composition'.

Line 47: 'composition' would be better as 'tissue composition'.

Line 50: 'in the same mouse in vivo' would be better if the authors added 'over time', or some longitudinal aspect of their study.

Line 75-76: 'feature extraction' would be better as 'feature extraction of the segmented nuclei'.

Line 97: 'right-shifted' might be better as 'red-shifted'.

Line 101-103: 'nuclei were manually selected from the pre-induction timepoint, wherever the green and red fluorescence signals overlapped in the post-induction timepoint images.' This phrase could be changed to: 'nuclei were manually selected from the pre-induction timepoint images (green fluorescence signal) wherever they overlapped the post-induction timepoint images (red fluorescence signal).'

Line 128-129: 'a single neural network classifier was trained for each cell type' would be better as 'five neural network classifiers were trained for each cell type'.

Lines 152-158: 'When the trained model was applied, two features not chosen by the automated feature selection algorithm differed significantly between the classes…' These two sentences are confusing here since it is not clear how they relate to the 'longitudinal' topic of this paragraph.

Line 182: 'z-step size' would be better as 'axial resolution'.

Line 183: 'more than 20000 cells' would be better as '~25000 cells' as stated earlier in the manuscript, or something similar.

Line 182-186: 'We were able to image a whole 3D volume of 700 μm³ consisting of more than 20000 cells in around 20 minutes, making it possible to perform large scale data acquisition in a repeated manner, enabling longitudinal intravital monitoring of tissue histology over time and in response to perturbations, treatments, or adaptive processes such as learning or exploring.' It is not clear whether the text following with 'making it possible to perform…' describes what the authors actually did in their study, or whether they are suggesting these methods are hypothetically possible. If it's the latter, then this sentence would be better if it was broken into two different sentences so there is no confusion.

Line 368: 'Nuclei touching the borders of the imaging volume were excluded.' The authors state in their reply that the error introduced by their counting method is small. They could include their reasoning here in the Methods, which would be helpful to others who decide to use the authors' methods.

Line 376: 'This allowed for identification of the label… ' It is not clear what label the authors are referring to here or why this label has not been identified. The previous mention of a 'unique label' indicates the labels were assigned manually and therefore should be known.

Line 406: "sequentialfs" should be "sequentials".

Figure 1B. There is a box for 'Trained network models' that is followed by what appears to be the trained network models. This looks redundant.

---

## [Editor Report · Decision Letter 3]

30 Sep 2023

Dear Dr Knabbe,

Thank you for the submission of your revised Methods and Resources Article "Accurate classification of major brain cell types using in vivo imaging and neural network processing" for publication in PLOS Biology. On behalf of my colleagues and the Academic Editor, Carole Parent, I am pleased to say that we can accept your manuscript for publication, provided you address any remaining formatting and reporting issues. These will be detailed in an email you should receive within 2-3 business days from our colleagues in the journal operations team; no action is required from you until then. Please note that we will not be able to formally accept your manuscript and schedule it for publication until you have completed any requested changes.

PRESS

Best wishes, 

Richard

Richard Hodge, PhD

rhodge@plos.org

PLOS
